# COA5 has an essential role in the early stage of mitochondrial complex IV assembly

Jia Xin Tang[1], Alfredo Cabrera-Orefice[2] , Jana Meisterknecht[2], Lucie S Taylor[1,3], Geoffray Monteuuis[4],
Maria Ekman Stensland[5], Adam Szczepanek[1], Karen Stals[6], James Davison[7,8], Langping He[3], Sila Hopton[3],
Tuula A Nyman[5] , Christopher B Jackson[4] , Angela Pyle[1], Monika Winter[1,9], Ilka Wittig[2], Robert W Taylor[1,3]

Pathogenic variants in cytochrome *c* oxidase assembly factor 5 (COA5), a proposed complex IV (CIV) assembly factor, have been shown to cause clinical mitochondrial disease with two siblings affected by neonatal hypertrophic cardiomyopathy manifesting a rare, homozygous *COA5* missense variant (NM_001008215.3: c.157G>C, p.Ala53Pro). The most striking observation in the affected individuals was an isolated impairment in the early stage of mitochondrial CIV assembly. In this study, we report an unrelated family in whom we have identified the same *COA5* variant with patient-derived fibroblasts and skeletal muscle biopsies replicating an isolated CIV deficiency. A CRISPR/Cas9-edited homozygous *COA5* knockout U2OS cell line with a similar biochemical profile was generated to interrogate the functional role of the human COA5 protein. Mitochondrial complexome profiling pinpointed a role of COA5 in early CIV assembly, more specifically, its involvement in the stage between MTCO1 maturation and the incorporation of MTCO2. We therefore propose that the COA5 protein plays an essential role in the biogenesis of MTCO2 and its integration into the early CIV assembly intermediate for downstream assembly of the functional holocomplex.

## Introduction

Mitochondria synthesise cellular energy in the form of adenosine triphosphate (ATP) via oxidative phosphorylation (OXPHOS), comprising four respiratory chain complexes and the $F_1F_O$ ATP synthase. Cytochrome *c* oxidase (COX), also known as complex IV (CIV), is the terminal electron acceptor of the respiratory chain, which catalyses the reduction of molecular oxygen to water. CIV couples this redox reaction to the translocation of protons across the inner mitochondrial membrane, thus contributing to the generation of the proton-motive force harnessed by the $F_1F_O$ ATP synthase to generate ATP.

CIV is comprised of 14 protein subunits of dual genetic origin; the three core subunits (MTCO1, MTCO2, and MTCO3) are all mitochondrially encoded, whereas the remaining subunits are encoded by the nuclear genome (Kadenbach, 2017; Wikström et al, 2018). Remarkably, CIV possesses two redox-active copper centres (binuclear $Cu_A$ and mononuclear $Cu_B$ centres) and two haem groups (haem *a* and haem $a_3$) (Wikström et al, 2018). These redox-active cofactors are crucial for electron transfer within CIV, which entails: (i) the receipt of electrons from reduced cytochrome *c* by the $Cu_A$ centre in the MTCO2 subunit, and (ii) subsequent delivery of the electron by the haem *a* group in a membrane-spanning MTCO1 subunit to (iii) the oxygen-reducing haem $a_3$-$Cu_B$ centre (Belevich et al, 2006; Muramoto et al, 2010; Kirchberg et al, 2012). These reactions result in the pumping of a total of four protons per oxygen molecule into the mitochondrial intermembrane space (IMS).

As a consequence, intricate assembly machinery has been generally described for complex IV on a modular basis centred around the three catalytic subunits: MTCO1, MTCO2, and MTCO3 with over 20 unique assembly factors of CIV having been characterised to date (Vidoni et al, 2017; Signes & Fernandez-Vizarra, 2018; Watson & McStay, 2020). These assembly factors are not only involved in the sequential incorporation of the protein subunits but also crucial for auxiliary processes such as translational regulation, protein stabilisation, and the insertion of cofactors and prosthetic groups (Watson & McStay, 2020; Povea-Cabello et al, 2024).

The cytochrome *c* oxidase assembly factor 5 (*COA5*) gene (RefSeq: NM_001008215.3), previously denoted as *C2orf64*, was first reported

[1]Mitochondrial Research Group, Translational and Clinical Research Institute, Faculty of Medical Sciences, Newcastle University, Newcastle upon Tyne, UK   [2]Functional Proteomics Center, Institute for Cardiovascular Physiology, Goethe University, Frankfurt am Main, Germany   [3]NHS Highly Specialised Rare Mitochondrial Disorders Service, Newcastle upon Tyne Hospitals, NHS Foundation Trust, Newcastle upon Tyne, UK   [4]Department of Biochemistry and Developmental Biology, Faculty of Medicine, University of Helsinki, Helsinki, Finland   [5]Department of Immunology, Institute of Clinical Medicine, University of Oslo and Oslo University Hospital, Oslo, Norway   [6]Department of Molecular Genetics, Royal Devon and Exeter NHS Foundation Trust, Exeter, UK   [7]Department of Paediatric Metabolic Medicine, Great Ormond Street Hospital for Children NHS Foundation Trust, London, UK   [8]National Institute of Health Research, Great Ormond Street Hospital Biomedical Research Centre, London, UK   [9]Department of Applied Sciences, Faculty of Health and Life Sciences, Northumbria University, Newcastle upon Tyne, UK

Correspondence: robert.taylor@ncl.ac.uk
Jia Xin Tang's present address is Department of NanoBiophotonics, Max Planck Institute for Multidisciplinary Sciences, Göttingen, Germany

in humans when a homozygous missense variant (c.157G>C, p.Ala53Pro) in this gene was shown to cause mitochondrial disease. Biochemical studies of patient-derived fibroblasts revealed isolated COX deficiency, more specifically the accumulation of CIV assembly intermediates and decreased levels of fully assembled CIV holocomplexes, leading the authors to hypothesise that COA5 is involved in the early stages of CIV assembly (Huigsloot et al, 2011). However, earlier studies carried out using the yeast orthologue of human COA5 protein, Pet191, also suggested a putative role in CIV assembly but with no impact on COX translation and copper metalation of the protein (McEwen et al, 1993; Tay et al, 2004; Khalimonchuk et al, 2008). Interestingly, contradicting evidence has been published with regard to the mitochondrial localisation of the Pet191 protein despite being a member of the twin $CX_9C$ protein family that are often found to be dependent on the Mitochondrial Intermembrane space Import and Assembly (MIA) pathway (Khalimonchuk et al, 2008; Bragoszewski et al, 2013).

Here, we present an unrelated family in which a clinically affected child harbours the identical *COA5* missense variant, identified by trio whole-exome sequencing (Longen et al, 2009). We generated a CRISPR/Cas9-mediated *COA5* knockout (*COA5*[KO]) cell line to elucidate the role of COA5 in CIV assembly and its implications on mitochondrial health and disease, using an array of biochemical tools including mitochondrial complexome profiling.

# Results

### Clinical summary and genomic studies

A female neonate born to second cousins of Turkish descent presented with tachypnoeic episodes on the first day of life. Blood lactate was elevated at variable levels between 2.5 and 11 mmol/litre. Clinical examination detected hyperdynamic precordium with loud second heart sound, and subsequent echocardiography indicated significant biventricular hypertrophic cardiomyopathy with septal hypertrophy and non-compaction appearance of the myocardium. Because of worsening respiratory distress, she was intubated and given inotropic support at 11 d after birth before being discharged at 6 wk. Liver function tests were abnormal with increased echogenicity of liver on ultrasound. Neuroimaging of the brain was normal. Focused metabolic biochemical investigations identified elevated urinary lactate and ethylmalonic acid. Clinical care was directed to symptomatic management only when a presumptive diagnosis of mitochondrial disorder with hypertrophic cardiomyopathy was made having identified evidence of significant complex IV activity deficiency on muscle biopsy as detailed below. A month later, she was readmitted with further deterioration of respiratory distress concurrent with a rhinovirus infection, poor cardiac function, and severe lactic acidosis, and finally passed away at 3 mo of age.

Molecular genetic testing eliminated pathogenic mitochondrial DNA (mtDNA) variants after a complete analysis of the mitochondrial genome. Trio whole-exome sequencing conducted at the Exeter Genomics Laboratory identified a previously reported *COA5* variant (c.157G>C, p.Ala53Pro), confirming both parents to be heterozygous carriers (Fig 1A) (Huigsloot et al, 2011; Chen et al, 2023). The homozygous *COA5* missense variant has been recorded on ClinVar as a pathogenic variant associated with isolated COX deficiency (https://www.ncbi.nlm.nih.gov/clinvar/variation/31087/). The c.157G>C *COA5* variant causes an amino acid change from alanine to proline at position 53 of the COA5 protein, which is located within the $CX_9C$ domain that has suggested linkage to mitochondrial protein localisation (Gladyck et al, 2021) (Fig 1B). The alanine residue is not conserved across species and is only shared between human and zebrafish (Fig 1B). Although the application of several in silico pathogenicity prediction tools suggested the c.157G>C *COA5* variant to be damaging, the REVEL meta-predictor score was below the recommended threshold for this variant, necessitating further study (Ioannidis et al, 2016).

### Patient-derived muscle biopsies and fibroblasts displayed isolated complex IV deficiency

Histochemical analysis of skeletal muscle cryosections from the patient indicated a moderate CIV deficiency in all fibres (Fig 2A). This was corroborated by quadruple OXPHOS immunofluorescence assay, which showed a marked loss of MTCO1 immunoreactivity (Fig 2B), whereas NDUFB8 (complex I subunit) protein levels were normal (Ahmed et al, 2017). The direct measurement of respiratory chain enzyme activities in patient-derived fibroblasts also indicated a severe and isolated complex IV deficiency in the *COA5* patient muscle sample (Fig 2C).

### Implications of the homozygous *COA5* variant on the steady-state level and assembly of complex IV

To characterise the pathogenicity of the *COA5* missense variant, SDS–PAGE and BN-PAGE were used to delineate its impact on OXPHOS protein steady-state levels and assembly. Decreased steady-state levels were only observed in the CIV subunit, MTCO2, whereas other OXPHOS complex subunits were unaffected in whole-cell lysates of both patient-derived fibroblasts and skeletal muscle biopsy (Fig 3A and C, quantification in Fig S1). Likewise for BN-PAGE, only complex IV assembly was impaired in patient-derived fibroblasts and skeletal muscle, whereas the remaining OXPHOS complexes were unaffected (Fig 3B and D, quantification in Fig S1). These observations were further supported by proteomic analysis of the immortalised patient fibroblasts, which highlighted the isolated CIV deficiency. CIV protein subunits exhibited a statistically significant decrease in protein abundance up to almost fourfold, whereas CI, CII, CIII, and CV protein abundance remained unchanged (Fig 3E, quantification in Fig S1).

### Functional characterisation of COA5 in a CRISPR/Cas9 knockout cell line

Both the primary and immortalised patient fibroblasts harbouring the homozygous c.157G>C *COA5* variant displayed arrested cell growth in culture, likely owing to the severe CIV deficiency and therefore making further experimentation challenging in primary fibroblasts. To enable in-depth characterisation of the functional role and the implicated pathogenicity of *COA5*, CRISPR/Cas9 gene

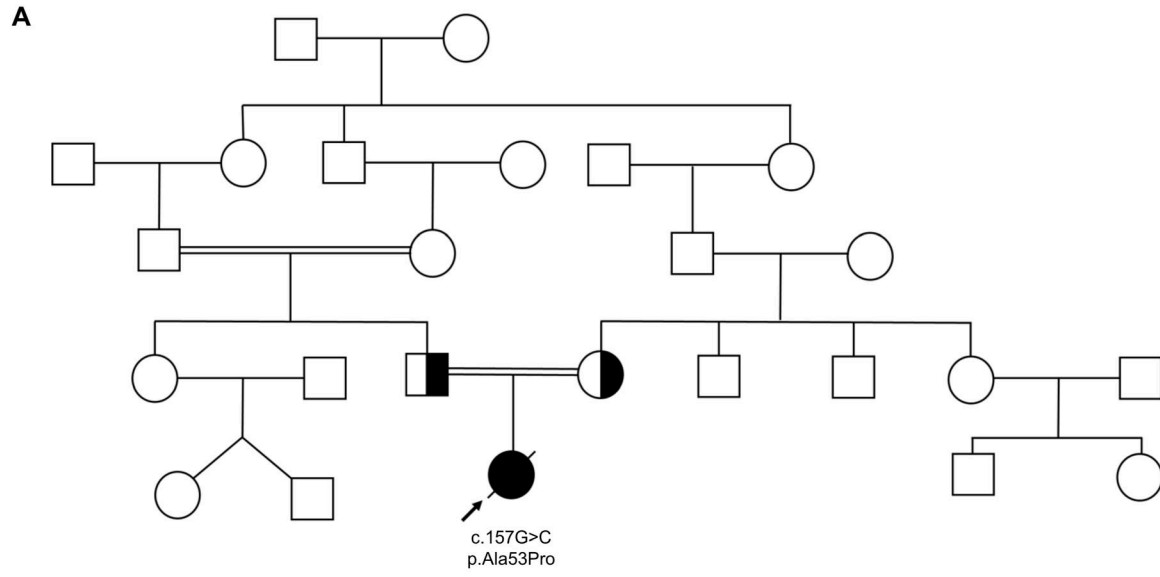

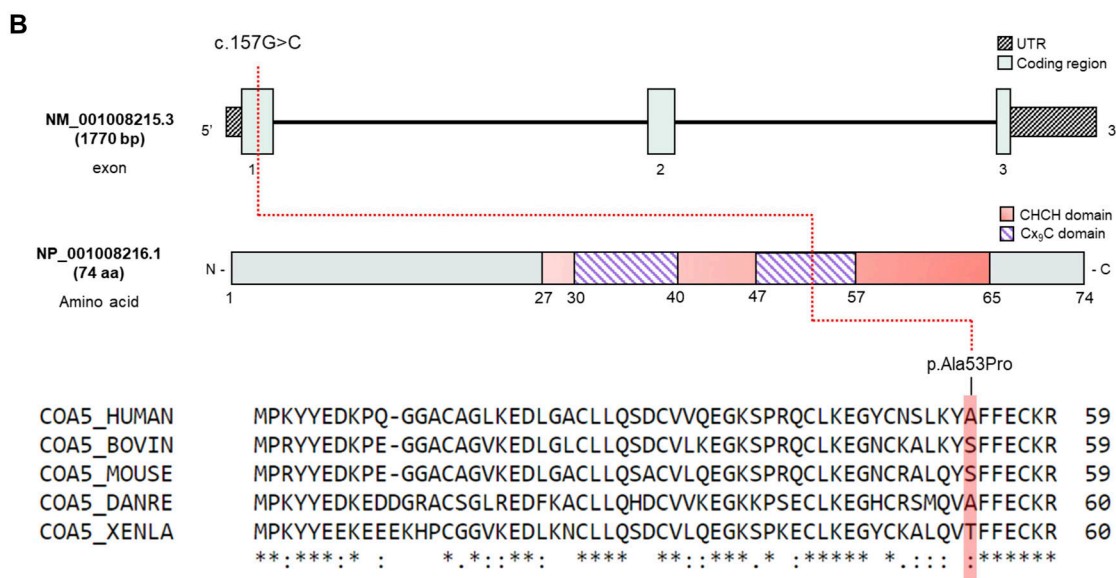

**Figure 1. Molecular genetics of the *COA5* variant.**
**(A)** Family pedigree of *COA5* patient showing segregation of the *COA5* variant. **(B)** Schematic representation of the COA5 gene (Ensembl) and its encoded protein (InterPro and UniProt) illustrating the nucleotide or amino acid impacted by the *COA5* variant. The coiled coil–helix–coiled coil helix (CHCH) domain containing twin CX₉C motifs (purple stripes) is illustrated in salmon pink. Amino acid residues at the position affected by the *COA5* variant across different species are highlighted.

editing was used to generate a homozygous *COA5* knockout (*COA5*^KO) in the immortalised human U2OS cell line for a more stable cellular model system. The *COA5*^KO cell line generated contained a homozygous seven base pair deletion in the *COA5* gene (c.287_290+3del, p.Val61del) as verified by Sanger sequencing (Fig S2). Western blot analyses of the *COA5*^KO cell line also confirmed an isolated CIV defect in terms of the protein steady-state level (Fig 4A and B, quantification in Fig S3) and OXPHOS complex assembly (Fig 4C, quantification in Fig S1), successfully mimicking the biochemical phenotype observed in patient-derived biopsies. Interestingly, an accumulation of the complex II subunit, SDHA protein, at around 70

kD was also observed on BN-PAGE analysis of the *COA5*^KO cell line, which was absent in the isogenic control (Fig 4C, quantification in Fig S1).

**Mitochondrial complexome profiling of the *COA5*^KO cell line**

To define the functional involvement of COA5 in CIV assembly, mitochondrial complexome profiling, which combines BN-PAGE and tandem mass spectrometry (LC-MS/MS), was used to determine the presence and arrangement of the OXPHOS system and related protein complexes (Alston et al, 2018; Alahmad et al, 2020;

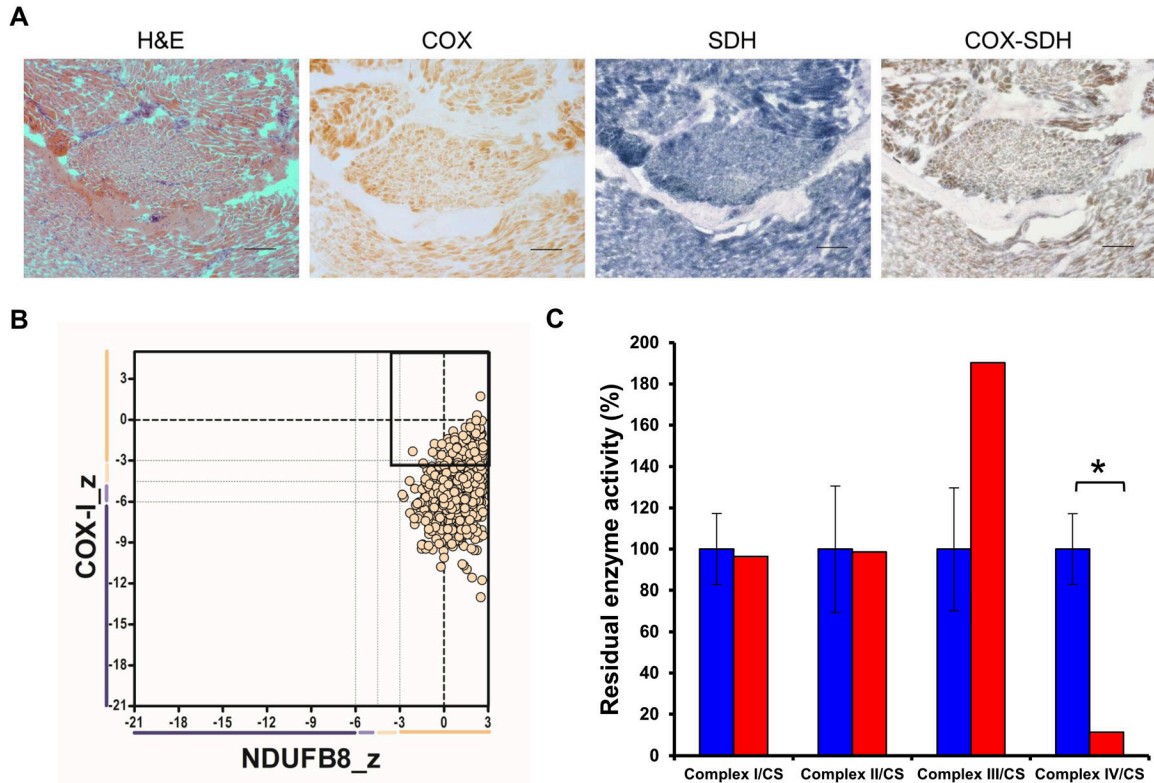

**Figure 2. Histopathology findings of patient with *COA5* variant.**
**(A)** Histochemical analysis of skeletal muscle section from a patient with haematoxylin and eosin (H&E), cytochrome *c* oxidase (COX), succinate dehydrogenase (SDH), and sequential COX-SDH histochemistry. Scale bar = 100 µm; magnification at 10x. **(B)** Quadruple oxidative phosphorylation immunofluorescent assay of single skeletal muscle fibres from the patient showing immunoreactivity against complex I subunit, NDUFB8 (x-axis), and complex IV subunit, MTCO1 (y-axis), normalised with porin expression as a mitochondrial mass marker. Each dot corresponds to a single muscle fibre, and the beige colour corresponds to normal mitochondrial mass. Muscle fibres with a z-score of less than −3 SD are considered deficient. Bold dashed lines represent the mean expression level in healthy muscle fibres. **(C)** Spectrophotometric measurement of oxidative phosphorylation enzyme activities in patient fibroblasts (in red) compared with mean activities in age-matched controls (in blue) shown as 100%. All measurements were normalised to citrate synthase (CS) activity. Error bars represent SDs of enzyme activities in control fibroblasts (*n = 8*). Respiratory chain enzyme activities in the patient that exceed the control range are marked with an asterisk (*).

Lobo-Jarne et al, 2020; Cabrera-Orefice et al, 2021). This technique enables visualisation of the specific stage of CIV assembly impacted when COA5 is absent using the *COA5*[KO] cell line and potentially uncovering interacting partners of the COA5 protein to elucidate its functional role.

First, complexome profiling (Fig 5) confirmed our result on BN-PAGE (Fig 4). An increased abundance of respiratory supercomplexes containing complexes I and III$_2$ (S$_0$: I+III$_2$) was observed in the *COA5*[KO] cell line (Fig 5, right panel). As observed in BN-PAGE analysis of the *COA5*[KO] mitochondrial extracts (Fig 4C), accumulation of the SDHA subunit of CII was also detected by complexome profiling at a molecular size of ~80–100 kD (native calibration of soluble complexes), but not in the isogenic control (highlighted in the orange box in Fig S4, right panel). Interestingly, SDHAF1 and SDHAF2 comigrate at the same range indicating an assembly intermediate (Fig S5, right panel, orange box). When assessing complex III subunits, a complete loss of the supercomplexes III$_2$+IV (S$_s$) was observed in the *COA5*[KO] cell line (Fig 5). Of all the OXPHOS complexes, only complex V content and assembly were unaffected (Fig 5).

Next, we had a closer look to complex IV subunits and assembly factors. We noted a complete loss of the COA5 protein as the

respective 12 kD protein was not detected in the knockout cell line but clearly present in the isogenic control as highlighted (Fig 6A, highlighted in yellow). More importantly, the accumulation of early CIV assembly intermediates (Fig 6A, right panel, green boxed subunits and assembly factors) and the loss of fully assembled CIV holocomplexes were observed in contrast to the isogenic control cell line (Fig 6A, left panel). Interestingly, subunits of the MTCO1 and MTCO2 modules accumulate in this intermediate, suggesting that despite the lack of subunits in the individual modules, further assembly is already taking place. This provides supporting evidence for the putative role of COA5 as an assembly factor in early stages of complex IV biogenesis and the isolated CIV defect resulting from the loss of COA5 protein as observed in biochemical studies of patient-derived cells and tissues (Fig 3).

## Discussion

This study identified the previously reported c.157G>C, p.Ala53Pro *COA5* variant in another family of Turkish descent. The isolated CIV

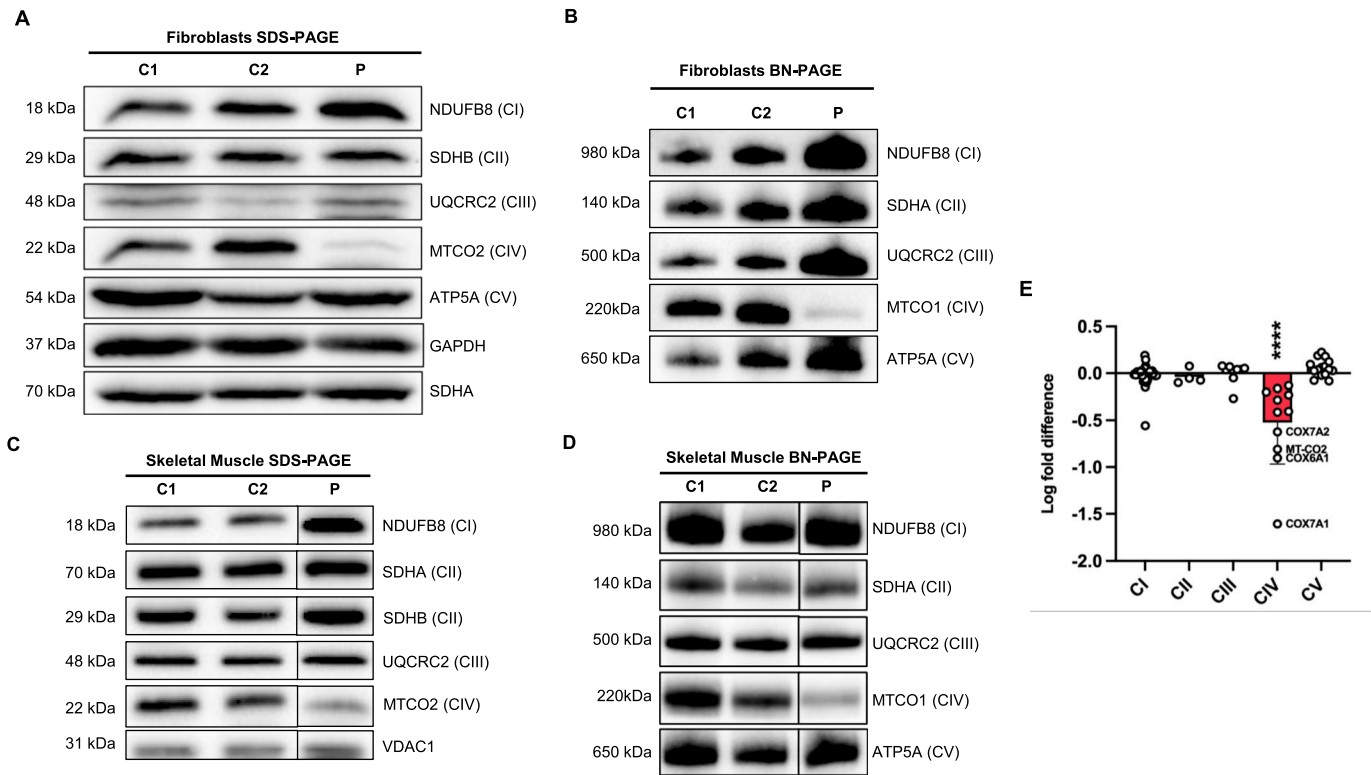

**Figure 3. Immunoblotting analyses of *COA5* patient-derived fibroblasts and skeletal muscle biopsies.**
**(A)** SDS–PAGE and immunoblotting analysis of whole protein lysates from *COA5* patient fibroblasts (P) and age-matched controls (C1 and C2) showing steady-state levels of oxidative phosphorylation (OXPHOS) complex subunits. GAPDH and SDHA were used as loading controls (*n* = 3). **(B)** BN-PAGE analysis of mitochondrial-enriched proteins solubilised in 2% DDM from the patient (P) against age-matched controls (C1 and C2) immunodetected against specific OXPHOS complex subunits. SDHA was used as a loading control (*n* = 3). **(C)** Western blot analysis of protein extracts from skeletal muscle sections of patient (P) and controls (C1 and C2) with VDAC1 as a loading control (*n* = 1). **(D)** BN-PAGE analysis of skeletal muscle mitochondrial extracts solubilised in 2% DDM derived from the patient (P) and 16-yr-old controls (C1 and C2) detecting all five OXPHOS complex subunits with SDHA as a loading control (*n* = 1). **(E)** Label-free proteomic profiling of immortalised patient fibroblasts with log fold change in abundance compared with control. Each data point corresponds to the mean of quadruplicate measurements of identified OXPHOS complex subunits (40 out of 44 CI subunits, 3 out of 4 CII subunits, 9 out of 10 CIII subunits, 11 out of 19 CIV subunits, and 15 out of 20 CV subunits were analysed).
Source data are available for this figure.

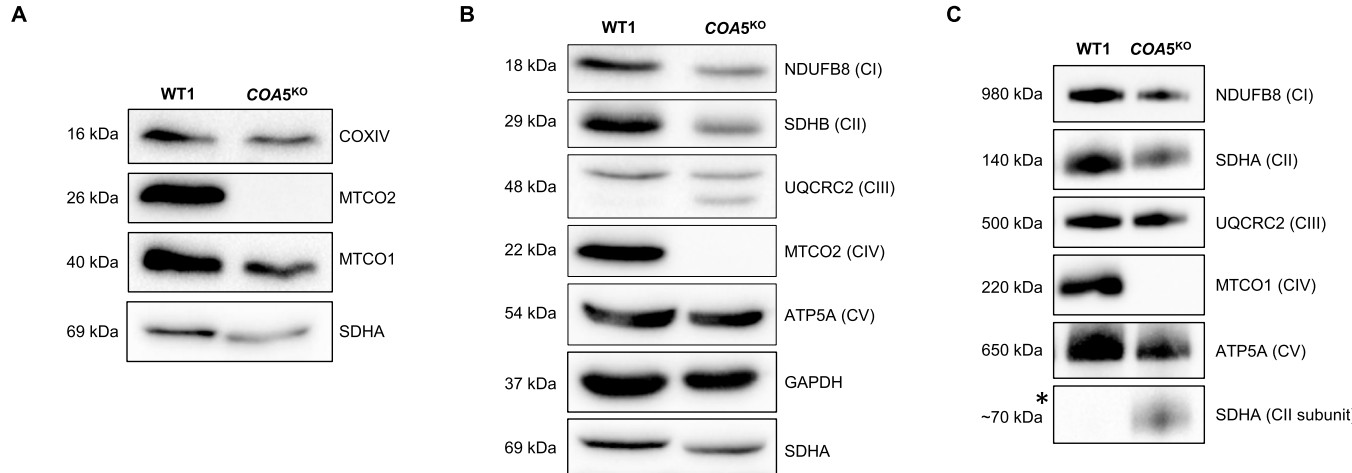

**Figure 4. Western blot analyses of the *COA5*^KO cell line against isogenic control.**
**(A, B)** SDS–PAGE of whole-cell lysates against (A) complex IV subunits including MTCO1, MTCO2, and COXIV (*n* = 2) and (B) protein subunits of all five oxidative phosphorylation (OXPHOS) complexes using OXPHOS cocktail antibody (*n* = 2). GAPDH and/or SDHA were used as loading controls. **(C)** BN-PAGE of mitochondrial-enriched lysates immunoblotted against all five OXPHOS complexes with SDHA as a loading control (*n* = 3).
Source data are available for this figure.

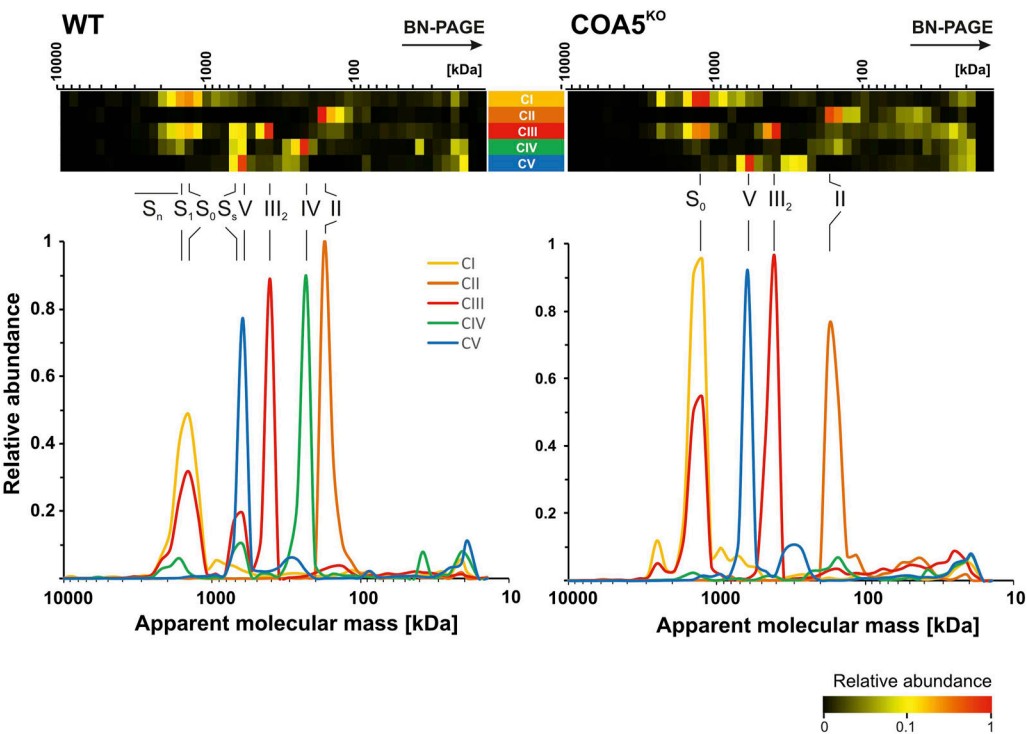

**Figure 5. Complexome profiling identified decreased complex IV levels in *COA5*^KO compared with the WT (*COA5*^WT) isogenic control cell line.**
Isolated mitochondrial membranes were solubilised with digitonin, and protein complexes were separated by BN-PAGE followed by quantitative mass spectrometric analysis. The quantitative data of identified individual subunits were summed up for each oxidative phosphorylation complex and normalised to maximum appearance between both cell lines. The data are presented as heatmaps and 2D plots, corresponding to protein components of individual oxidative phosphorylation complexes. The size of the complexes ranges from 10,000 to 10 kD (from left to right). The corresponding complexes were highlighted above the heatmap and 2D plots: fully assembled complex IV holocomplexes (IV), complex III dimer (III$_2$), supercomplexes of complex III dimers and complex IV (S$_S$), and supercomplexes containing complex I, complex III dimer (S$_0$), and complex IV (S$_1$).

deficiency associated with the *COA5* variant was observed in the patient fibroblasts and skeletal muscle biopsy, firmly supported by unbiased proteomic profiling of the immortalised *COA5* patient fibroblasts as the biochemical signature of COA5 deficiency. Most importantly, a CRISPR/Cas9 *COA5*^KO cell line enabled further interrogation via complexome profiling, narrowing down the involvement of COA5 protein to a specific stage of CIV biogenesis involving MTCO2 stabilisation and its incorporation into the MTCO1-containing subcomplex. This study pinpoints specific questions that arise with regard to the functional role of the COA5 protein as a CIV assembly factor and its impact on supercomplex formation, which we elaborate on further below.

Given that the rare c.157G>C, p.Ala53Pro variant was found only in two incidences where both patients were of Turkish ethnicity, this could point towards the likelihood of the variant being a founder pathogenic variant within the Turkish population. However, this would require further verification through haplotype analysis, necessitating access to patient-derived samples from the previous case. Importantly, overlapping clinical and biochemical phenotypes were observed between the proband in this study and the previously reported patient by Huigsloot and colleagues back in 2019 (Huigsloot et al, 2011), further strengthening the claim that the c.157G>C (p.Ala53Pro) *COA5* variant is pathogenic and causative of an isolated mitochondrial complex IV deficiency, associated with

loss of steady-state COX proteins and a COX assembly defect in isolation (Figs 2A–C and 3).

To further characterise the functional impact of COA5, we generated a CRISPR/Cas9-mediated knockout. The *COA5*^KO cells successfully replicated the isolated complex IV deficiency observed in the reported patients by indicating an evident loss of MTCO2 protein (Fig 4A). However, MTCO1 was detected at comparable levels to the isogenic control on SDS–PAGE (Fig 4A). This suggests the presence of stabilised MTCO1 subunits despite not being assembled into functional holocomplexes, corroborating observations from previous studies (Bourens et al, 2014; Bourens & Barrientos, 2017; Aich et al, 2018; Lobo-Jarne et al, 2020; Timón-Gómez et al, 2020). The loss of the fully assembled complex IV in *COA5*^KO cells was not only observed in BN-PAGE analysis but also shown in complexome profiling of the *COA5*^KO cells. Most strikingly, this has also been similarly observed in a two-dimensional BN-PAGE of a patient fibroblast cell line in the previous report, which demonstrated elevated levels of MTCO1 subcomplex but a marked decrease in complex IV holocomplex (Huigsloot et al, 2011). Despite this, the remaining OXPHOS complexes (complex I, II, III, and V) were unaffected except for the unusual accumulation of a protein complex of about 70 kD detected with SDHA, which corresponds to the size of the individual SDHA protein in the *COA5*^KO cell line.

To further infer the role of COA5 protein and its interacting partner, mitochondrial complexome profiling was employed to

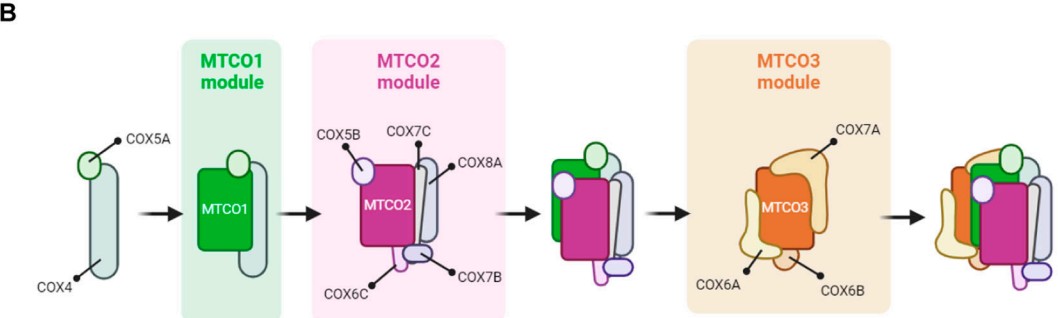

**Figure 6. Accumulation of early-stage complex IV assembly intermediate.**
**(A)** Following heatmap illustrates the distribution of complex IV subunits, assembly factors, the prohibitin complex (turquoise), complex I (yellow), and complex III (red). The subunits and assembly factors of the assembly modules are cultured as follows: MTCO1 module (green), MTCO2 module (purple), MTCO3 module (orange), and final subunit (black). The light blue arrows indicate subunits that comigrate with the prohibitin complex. **(B)** Schematic illustration of the sequential assembly intermediates of the CIV holocomplex. Adapted from Hock et al (2020).

dissect comigrating mitochondrial OXPHOS complexes in the COA5[KO] cell line in comparison with its isogenic control. The complexome data highlighted the sensitivity of the mass spectrometry–based approach in determining protein abundance as individual subunits and in higher molecular mass complexes,

enabling higher throughput and more robust quantification of changes in protein levels compared with SDS–PAGE and BN-PAGE. Notably, the knockout of COA5 protein was confirmed using this method despite the unavailability of a robust antibody for Western blotting. Most strikingly, complexome profiling of the COA5[KO] cell

line presented evidence of the implication of COA5 in early complex IV assembly, pinpointing its involvement to the stage between MTCO1 maturation and incorporation of MTCO2 into the early assembly intermediate (Fig 6B). This is supported by the following observations: the accumulation of MTCO1-containing sub-complexes (ranging from ~100 to 300 kD), which are absent in the isogenic control, (ii) the apparent absence of MTCO2 subunit, and (iii) the loss of fully assembled complex IV (Fig 6A).

Moreover, high molecular mass complexes containing MTCO1 and a number of nuclear-encoded CIV subunits were observed to colocalise with prohibitin complexes, which may point towards the likelihood of free complex IV subunits being directed for degradation because complex IV biogenesis was completely obstructed (Fig 6A). However, this phenomenon has only been described in yeast, which lacks complex I and therefore does not confer CI-containing supercomplexes, which also run at a similar, higher molecular mass (Steglich et al, 1999; Back et al, 2002; Kohler et al, 2023). A more likely explanation for the high molecular mass complexes containing MTCO1 submodules in the $COA5^{KO}$ cell line is the supercomplex $S_0$ (CI+III2). Notably, the association of MTCO1 and other nuclear-encoded CIV subunits with the supercomplex $S_0$ scaffold has been identified previously via co-immunoprecipitation and assembly kinetics analyses in the absence of MTCO2 (Lobo-Jarne et al, 2020; Timón-Gómez et al, 2020). The complexome profiling of $COA5^{KO}$ cell lines further strengthened our hypothesis that the MTCO1 submodule could assemble distinctively into supercomplexes, apart from its assembly into CIV holocomplexes, for the stabilisation of $S_0$ scaffold before supercomplex I+III$_2$+IV formation.

Furthermore, the higher abundance of complex I subunits, which was also observed in the SDS–PAGE analysis of $COA5^{KO}$ patient-derived lysates, suggests an accumulation of supercomplex $S_0$ (CI+III$_2$), which could not form a full respirasome in the absence of CIV holocomplexes. In addition, the complete loss of supercomplex III$_2$+IV reinforced the impact of $COA5^{KO}$ resulting in complex IV loss, hence favouring the formation of supercomplexes only containing complexes I and III$_2$ to mitigate the OXPHOS defect because of the absence of complex IV. Lastly, the accumulation of free SDHA was not seen in the isogenic control. This may either indicate an up-regulation of SDHA as a compensatory mechanism because of the OXPHOS defect or an accumulation of a comigrating complex of SDHA with CII assembly factors, such as SDHAF1 or SDHAF2 (Martínez-Reyes & Chandel, 2020).

In conclusion, this study provides insights into a distinct mode of pathological complex IV assembly caused by the assembly factor COA5, which specifically disrupts the transitional stage between MTCO1 maturation and MTCO2 incorporation. We present functional evidence to support a role of human COA5 protein in the early stage of complex IV assembly, corroborating its pathogenicity that can contribute to isolated complex IV deficiency. Interestingly, Nývltová et al suggested a role of COA5 in the stabilisation of what these authors termed "metallochaperone complexes," comprising COX-specific copper chaperones and haem biosynthesis enzymes required for the maturation and assembly of COX subunits (Nývltová et al, 2022). Although beyond the scope of this article, future experiments should focus on interrogating the submitochondrial localisation of the human COA5 protein to dissect its relevance in

MTCO2 stabilisation and incorporation during CIV assembly. Overall, this warrants further investigation to uncover the structural and functional role of COA5 with new insights into understanding its specific involvement in complex IV biogenesis.

# Materials and Methods

### Ethical statement

Written informed consent was obtained from the family for the use of patient samples in this study in accordance with the Declaration of Helsinki protocols and ethical approvals of local institutional review boards.

### Whole-exome sequencing

Trio WES (https://www.exeterlaboratory.com/genetics/genome-sequencing/) was conducted at the Exeter Genomics Laboratory as previously described (Chen et al, 2023).

### Histopathological and biochemical analyses

10 $\mu m$ of frozen skeletal muscle sections was used in each assessment. For histopathological studies, H&E staining was employed to determine muscle morphology, whereas sequential COX and SDH histochemistry was used to assess COX activity in muscle fibres. Spectrophotometric measurements of OXPHOS enzyme (complexes I-IV) activities were conducted as described in Taylor et al (2014) relative to citrate synthase activity. Quadruple immunofluorescence assays were carried out by measuring NDUFB8 (CI) and MTCO1 (Reddy et al, 2015) protein abundance against the mitochondrial mass marker, porin, using in-house analysis software as outlined in Taylor et al (2014).

### Cell culture

Patient- and age-matched control fibroblasts, as well as U2OS cells, were cultured in High Glucose DMEM supplemented with 10% FBS, 1X non-essential amino acids, 50 $\mu g/ml$ penicillin, 50 $\mu g/ml$ streptomycin, and 50 $\mu g/ml$ uridine.

### SDS–PAGE

Cell pellets were harvested when reaching 80–90% confluency and resuspended in lysis buffer (50 mM Tris–HCl [pH 7.5], 130 mM NaCl, 2 mM MgCl$_2$, 1 mM phenylmethanesulphonyl fluoride, 1% [vol/vol] Nonidet P-40, and 1X EDTA-free protease inhibitor cocktail). The resuspended cell pellets were incubated on ice for 10 min, before harvesting the resultant supernatant from centrifugation at 500$g$ at 4°C for 5 min, and the protein concentrations were determined using the Bradford assay.

Skeletal muscle homogenates were prepared by grinding 20 mg of frozen muscle section into powder using pestle and mortar in liquid nitrogen and resuspended in radioimmunoprecipitation (RIPA) buffer containing 1% IGEPAL, 1.5% Triton X-100, 0.5% sodium

deoxycholate, 10 mM β-mercaptoethanol, 0.1% SDS, 1 mM PMSF, and 1X EDTA-free protease inhibitor cocktail. The resuspensions were subjected to a 45-min incubation on ice followed by three rounds of 15-s homogenisation. The soluble proteins were extracted by centrifugation at 14,000$g$ at 4°C for 10 min, and protein concentrations were estimated using Pierce BCA Protein Assay Kit.

40 μg of protein extracts was resuspended in 1X Laemmli sample buffer and denatured at either 95°C for 5 min or 37°C for 15 min. The samples were then subjected to 12% SDS–PAGE with the Mini-PROTEAN Tetra Cell system and transferred onto a methanol-activated Immobilon-P polyvinylidene fluoride (PVDF) membrane using the Mini Trans-Blot Cell system.

### BN-PAGE

Cells were pelleted and resuspended in cell homogenisation buffer comprising 0.6 M mannitol, 1 mM EGTA, 10 mM Tris–HCl, pH 7.4, 1 mM PMSF, and 0.1% (vol/vol) BSA. The cell suspensions were subjected to three rounds of 15x homogenisation in Teflon–glass homogenisers at 4°C with intermittent differential centrifugation at 400$g$ for 10 min at 4°C to separate cytosolic protein fraction. The mitochondrial fractions were pelleted at 11,000$g$ for 10 min at 4°C and washed in cell homogenisation buffer without BSA.

Approximately 40 mg of skeletal muscle sections was processed and homogenised in muscle homogenisation buffer (250 mM sucrose, 20 mM imidazole hydrochloride, and 100 mM PMSF) and homogenised in Teflon–glass Dounce homogeniser for 15–20 rounds at 4°C. The muscle homogenates were then pelleted at 20,000$g$ for 10 min at 4°C and washed twice with muscle homogenisation buffer before pelleting at 20,000$g$ for 5 min at 4°C.

The final pellets were solubilised in 2% n-dodecyl-β-D-maltoside (DDM) and subjected to ultracentrifugation at 100,000$g$ for 15 min at 4°C, and the supernatants were extracted for protein concentration determination using Pierce BCA Protein Assay Kit. About 10 μg of mitochondrial protein complexes was loaded and separated in the precast Native PAGE 4–16% Bis-Tris 1.0 mm Mini Protein Gel in XCell SureLock Mini-Cell Electrophoresis System based on the manufacturer's protocol. The protein complexes were then immobilised onto an Immobilon-P PVDF membrane using the Mini Trans-Blot Cell system.

### Immunoblotting analysis

The membranes were blocked in 5% milk for an hour at room temperature before immunoblotting with specific primary antibodies and corresponding HRP-conjugated secondary antibodies as listed: OXPHOS cocktail (ab110411; Abcam), MTCO1 (ab14705; Abcam), MTCO2 (ab110258; Abcam), COXIV (ab14744; Abcam), NDUFB8 (ab110242; Abcam), SDHA (ab14715; Abcam), UQCRC2 (ab14745; Abcam), ATP5A (ab14748; Abcam), GAPDH (600004; ProteinTech), VDAC1 (ab14734; Abcam), polyclonal rabbit anti-mouse Ig/HRP (P0161; Dako), and polyclonal swine anti-rabbit Ig/HRP (P0399; Dako).

Finally, the resultant signal was detected using SuperSignal West Pico PLUS Chemiluminescent Substrate (Thermo Fisher Scientific) and analysed with ChemiDoc XRS+ Imaging Systems and ImageLab software (Bio-Rad). Densitometry analysis was conducted using ImageLab software (Bio-Rad). A $t$ test was performed on pairwise comparisons of the signal intensities normalised to loading control(s).

### On-bead precipitation and protein digestion

Cell pellets were lysed with 120 μl RIPA buffer containing protease and phosphate inhibitors. 10 μg of protein from all cell lysates was precipitated with 70% acetonitrile onto magnetic beads (MagReSyn Amine, Resyn Biosciences). The proteins were washed on the beads with 100% acetonitrile and 70% ethanol and then resuspended in 50 μl 50 mM ammonium bicarbonate containing 10 mM DTT for reduction of cysteines. Samples were incubated at 37°C for 40 min. Then, to alkylate proteins, 50 μl of 30 mM IAA in 50 mM ammonium bicarbonate was added and samples were incubated at RT in the dark for 30 min. 0.5 μg trypsin was added to each sample for overnight on-bead protein digestion at 37°C. The resulting peptides were concentrated and desalted on EVOTIPS for mass spectrometry analysis according to the standard protocol from EVOSEP.

### LC-MS/MS analysis

LC-MS/MS analysis was carried out using an EVOSEP one LC system (EVOSEP Biosystems) coupled to a timsTOF Pro2 mass spectrometer, using a CaptiveSpray nano-electrospray ion source (Bruker Corporation).

200 ng of digested peptides was loaded onto a capillary C18 column (15 cm length, 150 μm inner diameter, 1.5 μm particle size, EVOSEP). Peptides were separated at 40°C using the standard 30 sample/day method from EVOSEP.

The timsTOF Pro2 mass spectrometer was operated in DIA-PASEF mode. Mass spectra for MS were recorded between m/z 100 and 1700. Ion mobility resolution was set to 0.85–1.30 V·s/cm over a ramp time of 100 ms. The MS/MS mass range was limited to m/z 475–1,000 and ion mobility resolution to 0.85–1.27 V s/cm to exclude singly charged ions. The estimated cycle time was 0.95 s with 8 cycles using DIA windows of 25 D. Collisional energy was ramped from 20 eV at 0.60 V s/cm to 59 eV at 1.60 V s/cm.

Raw data files from LC-MS/MS analyses were submitted to DIA-NN (version 1.8.1) for protein identification and label-free quantification using the library-free function. The UniProt human database (UniProt Consortium, European Bioinformatics Institute, EMBL-EBI, UK) was used to generate library in silico from a human FASTA file. Carbamidomethyl (C) was set as a fixed modification. Trypsin without proline restriction enzyme option was used, with one allowed miscleavage, and peptide length range was set to 7–30 amino acids. The mass accuracy was set to 15 ppm, and the precursor false discovery rate (FDR) allowed was 0.01 (1%).

LC-MS/MS data quality evaluation and statistical analysis were done using software Perseus ver 1.6.15.0 CRISPR/Cas9 Gene Knockout.

WT U2OS cells were resuspended in room temperature Nucleofector Solution added with a supplement from the Cell Line Nucleofector Kit V (Lonza) at a density of 1 × 10$^6$ cells per nucleofection reaction. COA5-targeting sgRNA (Sigma-Aldrich) was incubated at room temperature with HiFi Cas9 nuclease at a 1:1.2 M ratio to form ribonucleoprotein complexes. The sgRNA sequence

designed to mediate CRISPR/Cas9 knockout of the *COA5* gene is as follows: 5'-TTTTGAGTGTAAAAGATCAG-3'. The sgRNA-Cas9 RNP complexes were nucleofected into WT U2OS cells on Nucleofector 2b Device (Lonza). Nucleofected cells were resuspended in growth media as outlined in the "Cell culture" method section before transferring to a six-well plate. The cells were incubated at 37°C, 5% CO2 for 48 h before isolating into single-cell clones using FACSAria Fusion Flow Cytometer (BD Biosciences) in four 96-well plates. Sanger sequencing chromatographs of the selected single-cell clones were analysed using Inference of CRISPR Edit (ICE) analysis (Synthego; https://ice.synthego.com/) to identify isogenic controls and *COA5* knockout cell line.

### Complexome profiling (BN-PAGE, trypsin digestion, LC-MS/MS, quantification)

Enriched mitochondrial proteins were extracted from *COA5*[KO] and isogenic control cell lines and solubilised with digitonin as described in Giese et al (2021). An equal amount of the solubilised mitochondrial protein extracts were subjected to a 3–18% acrylamide gradient gel (14 × 14 cm) for BN-PAGE as outlined in Wittig et al (2006). The gel was then stained with Coomassie blue, cut into equal fractions, and then transferred to 96-well filter plates. The gel fractions were then destained in 50 mM ammonium bicarbonate (ABC) followed by protein reduction using 10 mM DTT and alkylation in 20 mM iodoacetamide. Protein digestion was carried out in digestion solution (5 ng trypsin/$\mu$l in 50 mM ABC, 10% acetonitrile [ACN], 0.01% [wt/vol] ProteaseMAX surfactant [Promega], 1 mM CaCl$_2$) at 37°C for at least 12 h. After the recovery in the new 96-well plate, the peptides were dried in a SpeedVac (Thermo Fisher Scientific) and finally resuspended in 1% ACN and 0.5% formic acid. Nano-liquid chromatography and mass spectrometry (nanoLC/MS) was carried out on Thermo Fisher Scientific Q Exactive Plus equipped with an ultra-high-performance liquid chromatography unit Dionex UltiMate 3000 (Thermo Fisher Scientific) and Nanospray Flex Ion Source (Thermo Fisher Scientific). The MS data were analysed using MaxQuant software at default settings, and the recorded intensity-based absolute quantification (iBAQ) values were normalised to the isogenic control cell line.

## Data Availability

The mass spectrometry proteomics data for label-free whole-cell proteomics and complexome profiling produced in this study have been deposited to the ProteomeXchange Consortium via the PRIDE partner repository (Perez-Riverol et al, 2022) and assigned the dataset identifier PXD050891 and PXD053461, respectively.

## Supplementary Information

## Acknowledgements

RW Taylor is supported by the Wellcome Centre for Mitochondrial Research (203105/Z/16/Z), the Medical Research Council (MRC) International Centre for Genomic Medicine in Neuromuscular Diseases (MR/S005021/1), the UK NIHR Biomedical Research Centre in Age and Age-Related Diseases award to the Newcastle upon Tyne Hospitals NHS Foundation, the Lily Foundation, LifeArc, and the UK NHS Highly Specialised Service for Rare Mitochondrial Disorders. RW Taylor and M Winter are supported by the Pathology Society; RW Taylor, M Winter, and A Pyle are supported by Mito Foundation. JX Tang was supported by a PhD studentship from the Lily Foundation and a Newcastle University Overseas Research Studentship award. CB Jackson is supported by funding from the Academy of Finland (decision #336455), the Magnus Ehrnrooth Foundation, and the Jane and Aatos Erkko Foundation (#230004). The Proteomics Core Facility, University of Oslo/Oslo University Hospital, is supported by the Core Facilities programme of the South-Eastern Norway Regional Health Authority, and is a member of the National Network of Advanced Proteomics Infrastructure (NAPI), which is funded by the Research Council of Norway INFRASTRUKTUR-programme (project number: 295910). I Wittig is supported by the Deutsche Forschungsgemeinschaft (DFG): SFB1531-S01, project number 456687919, and WI 3728/3-1, project number 515944830.

### Author Contributions

JX Tang: data curation, formal analysis, and writing—original draft, review, and editing.
A Cabrera-Orefice: data curation, formal analysis, and writing—review and editing.
J Meisterknecht: data curation, formal analysis, and writing—review and editing.
LS Taylor: data curation, formal analysis, and writing—review and editing.
G Monteuuis: data curation, formal analysis, and writing—review and editing.
ME Stensland: data curation, formal analysis, and writing—review and editing.
A Szczepanek: data curation, formal analysis, and writing—review and editing.
K Stals: data curation, formal analysis, and writing—review and editing.
J Davison: clinical care of the family.
L He: data curation, formal analysis, and writing—review and editing.
S Hopton: data curation, formal analysis, and writing—review and editing.
TA Nyman: data curation, formal analysis, and writing—review and editing.
CB Jackson: supervision and writing—review and editing.
A Pyle: conceptualisation, supervision, funding acquisition, and writing—original draft, review, and editing.
M Winter: conceptualisation, supervision, funding acquisition, and writing—original draft, review, and editing.
I Wittig: conceptualisation, funding acquisition, and writing—review and editing.
RW Taylor: conceptualisation, supervision, funding acquisition, and writing—original draft, review, and editing.

### Conflict of Interest Statement

The authors declare that they have no conflict of interest.

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
