## [Reviewer comments · Life Science Alliance]

Life Science Alliance

COA5 has an essential role in the early stage of mitochondrial complex IV assembly

Jia Xin Tang, Alfredo Cabrera-Orefice, Jana Meisterknecht, Lucie Taylor, Geoffray Monteuuis, Maria Ekman Stensland, Adam Szczepanek, Karen Stals, James Davison, Langping He, Sila Hopton, Tuula Nyman, Christopher Jackson, Angela Pyle, Monika Winter, Ilka Wittig, and Robert Taylor

DOI: <https://doi.org/10.26508/lsa.202403013>

Corresponding author(s): Robert Taylor, Newcastle University

Review Timeline:

Submission Date:	2024-08-23
Editorial Decision:	2024-09-27
Revision Received:	2024-11-20
Editorial Decision:	2024-12-11
Revision Received:	2024-12-19
Accepted:	2024-12-19

Transaction Report:

September 27, 2024

Re: Life Science Alliance manuscript #LSA-2024-03013-T

Prof. Robert W Taylor
Newcastle University, Institute of Neuroscience
Wellcome Trust Centre for Mitochondrial Research
The Medical School
Framlington Place
Newcastle upon Tyne NE1 7RU
United Kingdom

Dear Dr. Taylor,

Thank you for submitting your manuscript entitled "COA5 has an essential role in the early stage of mitochondrial complex IV assembly" to Life Science Alliance. The manuscript was assessed by expert reviewers, whose comments are appended to this letter. We invite you to submit a revised manuscript addressing the Reviewer comments.

Thank you for this interesting contribution to Life Science Alliance. We are looking forward to receiving your revised manuscript.

Sincerely,

B. MANUSCRIPT ORGANIZATION AND FORMATTING:

Reviewer #1 (Comments to the Authors (Required)):

In this work, Tang et al. report a patient with complex IV (CIV) deficiency caused by a previously described pathogenic variant in the C2orf64 gene, encoding the CIV assembly factor COA5. The authors use biochemical techniques, proteomics and complexome profiling to define the role of COA5 in CIV biogenesis, concluding that it plays a critical role in the early stages of CIV assembly. This work is clearly written and technically sound. But there are contradictions between the western-blot and the CP data; and statistical analyses, necessary to prove the robustness of these experimental results, are missing. Additionally, a number of conceptual errors affect data presentation and interpretation, and technical aspects of this work must be improved in order to reach the right conclusions.

Major points:

- Despite Huigsloot et al. (2011) previously described COA5 to affect the initial steps of CIV assembly, according to the data presented here COA5 clearly plays a role in the intermediate stages of CIV assembly, rather than in the initial ones. The first step involves mainly the maturation of MT-CO1 and its binding to the COX4-COX5A submodule. The second step (intermediate) involves the maturation of MT-CO2 and its binding to the initial MT-CO1-COX4-COX5A module. The final step involves the formation and binding of the MT-CO3 module to complete the assembly of the holoenzyme. COA5 loss mainly affects the stability of MT-CO2 and of the subunits of the MT-CO3 module (Fig3E) without massively affecting the MT-CO1 module; so, it participates in the intermediate stage. Therefore: i) indicate these steps clearly in the CIV assembly model shown in Fig6B and ii) do not mix up the different stages of this process, correct "early assembly" by "intermediate assembly" all throughout the manuscript.
- Fig2C. Indicate the number of experimental replicates done with the patient's samples, add error bars and make proper statistical calculations. Is the increase in CIII activity reproducible and does it fit with an increase in CIII protein levels? According to CP, yes. According to Western-blot, calculations are needed.
- Fig3: The figure should be self-explicative. Indicate on top of each panel the sample type (fibros, muscle) and the experimental approach (SDS-PAGE, BN-PAGE) to facilitate reading. Indicate in the figure legend the solubilisation conditions used in fig3B,D.
- Page 7 and Fig4: "Western blot analyses of the COA5KO cell line also confirmed an isolated CIV defect in terms of protein steady-state level (Fig 4A-B) as well as OXPHOS complex assembly (Fig 4C)". This is not so clear, as in figures 4A-B the signals for CI, CII, CIII, GAPDH, CV, etc are decreased in the COA5KO cell line and similarly, in fig4C all complexes go down except CII. The authors' claim will only be confirmed after proper quantification of the signals of all western blots from SDS-PAGE and BN-PAGE analyses, which should be normalized by a clearly unaltered protein/complex (maybe CV?) and plotted in a graph with valid statistical analyses. This is particularly important to validate the robustness of experimental data, given the apparent contradictory results between the SDS-PAGE, BN-PAGE and CP analyses.
- Fig4C: Why is CII is not properly detected in the wild-type cells, as it should be easily visualized? In fact, why are CII levels decreased upon COA5 loss in all the other samples/experiments, including the CP, but not in fig4C? Data should be coherent. It is difficult to imagine that CII levels increase in the mutant after seeing its levels decreased in CP, together with the accumulation of CII assembly intermediates. The apparent increase of CII levels in fig4C looks like the result of a solubilisation error in control cells when performing this particular BNE, in which CII would have shifted in size towards a higher/lower position. Alternatively, the selected exposure of the blots is too low. Irrespectively, the CII band for the control cells should be shown. Please avoid unnecessary speculation in this regard.
- Fig6A: Put all CIV assembly factors together, as it is difficult to understand why they are separated in two panels. HIGD2A has been already shown as part of the CIV intermediate that accumulates in the absence of MT-CO2 (doi: 10.15252/emj.2019103912 and doi: 10.1016/j.celrep.2020.107607), so place it next to COX15 inside the green square. Besides the PHB complexes, add in the lowest panel the positions of the average CI and CIII signals to show that the supercomplex S0 also co-aligns with several CIV subunits in the COA5 KO mutant. This is relevant for further discussion (below).

-Discussion, lines 233-236: "While spontaneous degradation of MTCO1 proteins which are unable to be associated with MTCO2 is normally expected, the unaffected steady-state levels of MTCO1 suggest the presence of stabilised MTCO1 subunits despite not being assembled into functional holocomplexes." This wrong statement should be corrected. In fact, there are multiple published papers showing that MT-CO1 is stabilized in the absence of MT-CO2 (dois: 10.15252/embj.2019103912; 10.1016/j.celrep.2020.107607; 10.1074/jbc.M117.778514; 10.7554/eLife.32572 and 10.1093/hmg/ddu003, to mention a few).

-Discussion, lines 254-260: "Notably, higher molecular mass complexes were shown containing complex IV subunits in COA5KO and could reflect assembly of intermediates with the supercomplex S0 scaffold (Fernández-Vizarrá & Ugalde, 2022). However, these are more likely to be prohibitin complexes, which direct the free complex IV subunits for degradation rather than supercomplex formation since complex IV biogenesis was completely obstructed (Fig 6A) (Back et al, 2002; Kohler et al, 2023; Steglich et al, 1999)."

In this manuscript the authors show MT-CO1 and a number of nuclear-encoded CIV subunits (COX4, COX5B, COX7A2L, COX7C, COX8A), exactly the same ones that were previously reported in two independent papers (please add them to the reference list) to be attached to the supercomplex S0 scaffold in human cells depleted of MT-CO2 (doi: 10.15252/embj.2019103912 and doi: 10.1016/j.celrep.2020.107607). This direct interaction of MT-CO1 and nuclear-encoded CIV subunits with the supercomplex S0 scaffold was demonstrated previously by co-immunoprecipitation and assembly kinetics analyses, and this binding stabilized CI activity and CI protein/complex levels when compared to cells without MT-CO1 (doi: 10.15252/embj.2019103912). Contrary to the authors' statements, the accumulation of nuclear-encoded CIV subunits in the prohibitin (PBH) complex is highly questionable: i) co-localization does not mean interaction; ii) why would these nuclear-encoded CIV proteins be imported in the mitochondria in the first place, then bound and stabilized at detectable levels in the PBH structures, if they are prone to degradation? It makes no sense; iii) the three papers provided by the authors to support their statements are based on work solely performed in yeast, which does not contain complex I (CI) and does not form CI-containing supercomplexes. None of these papers show clear evidence of interaction between nuclear-encoded CIV subunits and prohibitins, but (Kohler et al, 2023) instead supports the interaction of the yeast PHB/m-AAAProtease complex with newly-synthesized mitochondrially-encoded proteins to drive either their assembly in OXPHOS complexes or their degradation. The authors should avoid speculation based on unproven findings and low the tone regarding the likelihood of interaction with the PBH complex.

-Discussion, general comment: This section is way too long in its current form and could be easily shortened to a maximum of three pages. There is much irrelevant speculation about the hypothetical roles of COA5 according to its mitochondrial localization that is unrelated to the experimental evidence provided in this manuscript. The authors should focus on the role of COA5 on CIV assembly and how it affects the overall organization and function of the respiratory chain. For instance, why is CIII activity increased (fig2C) and how is it related to the increase of free CIII and SC I+III2 in the COA5 mutants?

Minor points:

- Lines 191-193: "When assessing complex III subunits, a complete loss of the supercomplexes III2+IV (S0) was observed in the COA5KO cell line (Fig 5)." SS, not S0, refers to the supercomplex III2+IV in fig5.

Reviewer #2 (Comments to the Authors (Required)):

In this manuscript, the authors present a case of a patient with a single-point mutation in the COA5 (PET191) gene, which was previously reported in two siblings from a Turkish family (2011). Their findings corroborate earlier studies (Huigsloot et al., 2011; Nyvltova et al., 2022), showing that COA5 is an essential assembly factor for complex IV (CIV) in humans. They confirm that, in the absence of COA5, early assembly intermediates of CIV accumulate. In general, the present study lacks novelty, which prevents support for its publication

Specific points:

1. The authors speculate that the mutation found in patients may affect the mitochondrial localization of the COA5 protein, but they have not provided evidence to support this. To address this question, it would be beneficial to express the mutant COA5 in a KO cell line; however, this experiment was not conducted.
2. In Figure 2C, the measurement of activities is missing standard deviations (SD) for patient fibroblasts. Additionally, while it appears that complex III (CIII) is increased, the statistical significance is not indicated. Please include a notation for non-significant results and specify the significance of the other findings.
3. In Figures 3A and 3C, the Western blot analysis of patient samples shows significant differences only in CIV subunits. However, only MTCO2 was investigated. Quantification is needed, as the figures also suggest changes in a CI subunit (NDUFB8) and in UQCRC2 (increased in one instance and decreased in another).
4. For the U2OS KO cell line, the authors should provide evidence that reintroducing the COA5 gene restores the KO phenotype. Additionally, they should demonstrate the absence of COA5 protein via WB, assuming the antibody is available and functional.
5. In Figure 4B, the authors should explain the additional band observed in UQCRC2 labeling in the KO cells.

6. Figures 4A, 4B, and 4C require better loading controls and quantification to assess the phenotype accurately. Suitable controls include Actin, VDAC, or TOM20.
7. Figure 5 presents complexome profiling data that appear inconsistent with earlier statements about changes in steady-state levels of CIV subunits. The authors claim differences in I + III2 supercomplexes, but quantification comparing WT and KO is needed to assess increased or decreased levels accurately.
8. The authors assert that COX1 steady-state levels are unaffected, yet no supporting evidence (e.g., WB) is provided.
9. The discussion section repeats well-established observations about the stability of COX1 and COX2 in subcomplexes, the assembly of CIV, and the absence of respirasomes without CIV. Furthermore, the investigation of COA5 as a potential copper chaperone has already been covered in previous research published by other groups.

Reviewer #3 (Comments to the Authors (Required)):

LSA-2024-03013-T

Tang et al. COA5 has an essential role in the early stage of mitochondrial complex IV assembly

The manuscript by Tang et al. studies a patient case with severe neonatal-onset metabolic disease and cardiomyopathy. Exome sequencing identified a previously described homozygous missense variant in the COA5 gene, which has been suggested to encode a complex IV assembly factor with thus far unknown mechanism of action. The authors utilize patient fibroblasts and muscle biopsy to show isolated CIV deficiency. They then go on to study CIV assembly in the fibroblasts and muscle biopsy using Western blot and BN-PAGE, which show a specific assembly defect. To aid the studies of CIV assembly, the authors generate a COA5 knock-out cell line using CRISPR/Cas9 and utilize this model to do mass spectrometric profiling of OXPHOS complexes, which further elaborates the role of COA5. The manuscript is well written, technically credible and all conclusion are supported by the experimental data. The only part that I find a bit puzzling is the claim that primary and immortalized patient fibroblasts grew poorly, and therefore generation of a knock-out cancer cell line was necessary. This is quite surprising because e.g. mtDNA-less rho zero cells can grow without the entire respiratory chain when uridine is supplemented. How were the fibroblasts immortalized? The patient cells harbor a COA5 missense mutation but the genome-edited cell line carries a full knock-out allele, yet the authors do not mention if it has a growth defect. The authors should either explain/clarify this and/or provide growth curves for the knock-out and control cell lines.

Dr. Jukka Kallijärvi

Life Science Alliance LSA-2024-03013-T**Reviewer #1 (Comments to the Authors (Required)):**

In this work, Tang et al. report a patient with complex IV (CIV) deficiency caused by a previously described pathogenic variant in the *C2orf64* gene, encoding the CIV assembly factor COA5. The authors use biochemical techniques, proteomics and complexome profiling to define the role of COA5 in CIV biogenesis, concluding that it plays a critical role in the early stages of CIV assembly. This work is clearly written and technically sound. But there are contradictions between the western-blot and the CP data; and statistical analyses, necessary to prove the robustness of these experimental results, are missing. Additionally, a number of conceptual errors affect data presentation and interpretation, and technical aspects of this work must be improved in order to reach the right conclusions.

We are grateful to this reviewer for their thorough evaluation of our manuscript and have tried to address all of their comments below.

Major points:

- Despite Huigsloot et al. (2011) previously described COA5 to affect the initial steps of CIV assembly, according to the data presented here COA5 clearly plays a role in the intermediate stages of CIV assembly, rather than in the initial ones. The first step involves mainly the maturation of MT-CO1 and its binding to the COX4-COX5A submodule. The second step (intermediate) involves the maturation of MT-CO2 and its binding to the initial MT-CO1-COX4-COX5A module. The final step involves the formation and binding of the MT-CO3 module to complete the assembly of the holoenzyme. COA5 loss mainly affects the stability of MT-CO2 and of the subunits of the MT-CO3 module (Fig3E) without massively affecting the MT-CO1 module; so, it participates in the intermediate stage. Therefore: i) indicate these steps clearly in the CIV assembly model shown in Fig6B and ii) do not mix up the different stages of this process, correct "early assembly" by "intermediate assembly" all throughout the manuscript.

We thank the reviewer for their insight and suggestions. The data presented in this manuscript corroborated the findings in Huigsloot et al. (2011) showing stalling of CIV assembly resulting in MTCO1 accumulation, suggesting impaired incorporation of MTCO2. Hence, the parallel biogenesis and incorporation of MTCO1 and MTCO2 are phrased as 'early stage' before they form an intermediate complex for the final maturation of the holocomplex with the incorporation of MT-CO3. This has also been similarly illustrated in the following references: PMID: 30030361 and 29381136.

- Fig2C. Indicate the number of experimental replicates done with the patient's samples, add error bars and make proper statistical calculations. Is the increase in CIII activity reproducible and does it fit with an increase in CIII protein levels? According to CP, yes. According to Western-blot, calculations are needed.

The assessment of OXPHOS enzyme activities in patient fibroblasts was only conducted once (activities are measured in triplicate, with different amounts of mitochondrial protein, and data presented as the mean of these three measurements) as part of an established diagnostic protocol (as exemplified in PMID: 31866046, 32969598) and therefore statistical significance could not be determined. This is now clearly indicated in the legend of Fig. 2C.

The increase in CIII enzyme activity, is believed to be a compensatory phenotype, which has been consistently observed in patients harbouring other mitochondrial disease-causing variants in genes including *OXA1L* (PMID: 30201738), *NDUFC2* (PMID: 32969598), *PET100* (PMID: 25293719) and *GFM2* (PMID: 29075935). Based on the quantification of the

corresponding BN-PAGE, the CIII protein levels were also significantly higher in patient fibroblasts compared to age-matched controls, fitting nicely with the increase in CIII activity (**Figure EV1B**).

- Fig3: The figure should be self-explicative. Indicate on top of each panel the sample type (fibros, muscle) and the experimental approach (SDS-PAGE, BN-PAGE) to facilitate reading. Indicate in the figure legend the solubilisation conditions used in fig3B,D.

We thank the reviewer for highlighting this. The appropriate labels have now been added to the respective panels in **Figure 3** and the solubilisation conditions used are now clearly stated in the Figure legend.

- Page 7 and Fig4: "Western blot analyses of the COA5KO cell line also confirmed an isolated CIV defect in terms of protein steady-state level (Fig 4A-B) as well as OXPHOS complex assembly (Fig 4C)". This is not so clear, as in figures 4A-B the signals for CI, CII, CIII, GAPDH, CV, etc are decreased in the COA5KO cell line and similarly, in fig4C all complexes go down except CII. The authors' claim will only be confirmed after proper quantification of the signals of all western blots from SDS-PAGE and BN-PAGE analyses, which should be normalized by a clearly unaltered protein/complex (maybe CV?) and plotted in a graph with valid statistical analyses. This is particularly important to validate the robustness of experimental data, given the apparent contradictory results between the SDS-PAGE, BN-PAGE and CP analyses.

Thank you for this comment. The quantification of western blot data are now included in **Fig EV1** and **Fig EV3**. We would like to highlight the higher sensitivity of complexome profiling compared to other biochemical characterisation approaches (SDS-PAGE and BN-PAGE), hence the complexomic data provides the most robust verification in terms of protein abundance quantification in individual subunits as well as in complexes.

- Fig4C: Why is CII is not properly detected in the wild-type cells, as it should be easily visualized? In fact, why are CII levels decreased upon COA5 loss in all the other samples/experiments, including the CP, but not in fig4C? Data should be coherent. It is difficult to imagine that CII levels increase in the mutant after seeing its levels decreased in CP, together with the accumulation of CII assembly intermediates. The apparent increase of CII levels in fig4C looks like the result of a solubilisation error in control cells when performing this particular BNE, in which CII would have shifted in size towards a higher/lower position. Alternatively, the selected exposure of the blots is too low. Irrespectively, the CII band for the control cells should be shown. Please avoid unnecessary speculation in this regard.

Thank you for highlighting this. Fully assembled CII was detected in both wild-type and COA5^{KO} cell lines at 140 kDa as highlighted in green (**Figure R1, see below**). It was also shown to be decreased in COA5^{KO} in comparison to the wildtype cells. The lowest panel highlighted in red (**Figure R1**), corresponding to SDHA subunit of CII (predicted based on size), was indeed only detected in the COA5^{KO} cell line. As this is likely to be an assembly intermediate of CII, we would not have anticipated to see this in the BN-PAGE of wildtype cells without an OXPHOS assembly defect, where protein complexes are preserved in their native states.

Figure R1 Corresponding to Fig 4C in the manuscript. Green box indicate fully assembled CII while red box highlights accumulation of unassembled SDHA subunit of CII in knockout cell line.

Both panels aligned with the findings in complexome profiling (Fig 5) where:

- (i) CII assembly intermediates (~70 kDa) were found to accumulate in COA5^{KO} cells but were absent in wildtype cells.
- (ii) A decreased level of fully assembled CII (140 kDa) was observed in COA5^{KO} cells compared to WT.

Furthermore, these observations were consistently replicated following triplicate repeats of these experiments (Fig R2, see below), eliminating – in our view – the possibility of solubilisation error.

Figure R2 Biological triplicates of Figure 4C with red arrows indicating the detected CII assembly intermediate likely containing SDHA subunit only in the COA5^{KO}-derived mitochondrial protein extracts. Fully assembled CII were detected in both isogenic control (WT) and knockout cell line as indicated in green arrows.

We apologise for any confusion this might have caused in our original submission and have now added an asterisk to the 70 kDa panel of CII assembly intermediate in Figure 4C to indicate the additional panel included to highlight this.

-Fig6A: Put all CIV assembly factors together, as it is difficult to understand why they are separated in two panels. HIGD2A has been already shown as part of the CIV intermediate that accumulates in the absence of MT-CO2 (doi: 10.15252/embj.2019103912 and doi: 10.1016/j.celrep.2020.107607), so place it next to COX15 inside the green square. Besides the PHB complexes, add in the lowest panel the positions of the average CI and CIII signals to show that the supercomplex S0 also co-aligns with several CIV subunits in the COA5 KO mutant. This is relevant for further discussion (below).

We thank the reviewer for pointing this out. The updated **Figure 6A** now combined the CIV assembly factors in one panel. HIG2DA has also been shifted next to COX15 and adapted into the green box whereas the averaged complex I and III protein abundance are now indicated at the bottom.

-Discussion, lines 233-236: "While spontaneous degradation of MTCO1 proteins which are unable to be associated with MTCO2 is normally expected, the unaffected steady-state levels of MTCO1 suggest the presence of stabilised MTCO1 subunits despite not being assembled into functional holocomplexes." This wrong statement should be corrected. In fact, there are multiple published papers showing that MT-CO1 is stabilized in the absence of MT-CO2 (dois: 10.15252/embj.2019103912; 10.1016/j.celrep.2020.107607; 10.1074/jbc.M117.778514; 10.7554/eLife.32572 and 10.1093/hmg/ddu003, to mention a few).

We thank the reviewer for pointing this out, and have now corrected this statement, adding the relevant references in lines 232-235.

-Discussion, lines 254-260: "Notably, higher molecular mass complexes were shown containing complex IV subunits in COA5KO and could reflect assembly of intermediates with the supercomplex S0 scaffold (Fernández-Vizorra & Ugalde, 2022). However, these are more likely to be prohibitin complexes, which direct the free complex IV subunits for degradation rather than supercomplex formation since complex IV biogenesis was completely obstructed (Fig 6A) (Back et al, 2002; Kohler et al, 2023; Steglich et al, 1999)."

In this manuscript the authors show MT-CO1 and a number of nuclear-encoded CIV subunits (COX4, COX5B, COX7A2L, COX7C, COX8A), exactly the same ones that were previously reported in two independent papers (please add them to the reference list) to be attached to the supercomplex S0 scaffold in human cells depleted of MT-CO2 (doi: 10.15252/embj.2019103912 and doi: 10.1016/j.celrep.2020.107607). This direct interaction of MT-CO1 and nuclear-encoded CIV subunits with the supercomplex S0 scaffold was demonstrated previously by co-immunoprecipitation and assembly kinetics analyses, and this binding stabilized CI activity and CI protein/complex levels when compared to cells without MT-CO1 (doi: 10.15252/embj.2019103912).

Contrary to the authors' statements, the accumulation of nuclear-encoded CIV subunits in the prohibitin (PBH) complex is highly questionable: i) co-localization does not mean interaction; ii) why would these nuclear-encoded CIV proteins be imported in the mitochondria in the first place, then bound and stabilized at detectable levels in the PBH structures, if they are prone to degradation? It makes no sense; iii) the three papers provided by the authors to support their statements are based on work solely performed in yeast, which does not contain complex I (CI) and does not form CI-containing supercomplexes. None of these papers show clear evidence of interaction between nuclear-encoded CIV subunits and prohibitins, but (Kohler et al, 2023) instead supports the interaction of the yeast PHB/m-AAAProtease complex with newly-synthesized mitochondrially-encoded proteins to drive either their assembly in OXPHOS complexes or their degradation. The authors should

avoid speculation based on unproven findings and low the tone regarding the likelihood of interaction with the PBH complex.

We thank the reviewer for their valuable input. The interaction of MTCO1 and nuclear-encoded CIV subunits to the S_0 has now been highlighted in the manuscript with the correct references included, and the interaction with prohibitin has been rephrased in lines 254-268.

-Discussion, general comment: This section is way too long in its current form and could be easily shortened to a maximum of three pages. There is much irrelevant speculation about the hypothetical roles of COA5 according to its mitochondrial localization that is unrelated to the experimental evidence provided in this manuscript. The authors should focus on the role of COA5 on CIV assembly and how it affects the overall organization and function of the respiratory chain. For instance, why is CIII activity increased (fig2C) and how is it related to the increase of free CIII and SC I+III2 in the COA5 mutants?

We thank the reviewer for their helpful comments. In line with these, we have restructured the discussion to focus on the role of COA5 in CIV assembly and its link to the function and organisation of respiratory chain complexes and supercomplexes.

Minor points:

- Lines 191-193: "When assessing complex III subunits, a complete loss of the supercomplexes III2+IV (S_0) was observed in the COA5KO cell line (Fig 5)." S_0 , not S_0 , refers to the supercomplex III2+IV in fig5.

This has now been corrected in the manuscript.

Reviewer #2 (Comments to the Authors (Required)):

In this manuscript, the authors present a case of a patient with a single-point mutation in the COA5 (PET191) gene, which was previously reported in two siblings from a Turkish family (2011). Their findings corroborate earlier studies (Huigsloot et al., 2011; Nyvltova et al., 2022), showing that COA5 is an essential assembly factor for complex IV (CIV) in humans. They confirm that, in the absence of COA5, early assembly intermediates of CIV accumulate. In general, the present study lacks novelty, which prevents support for its publication

We thank the reviewer for their helpful comments and are grateful for the opportunity to respond to all of the comments raised below.

Specific points:

1. The authors speculate that the mutation found in patients may affect the mitochondrial localization of the COA5 protein, but they have not provided evidence to support this. To address this question, it would be beneficial to express the mutant COA5 in a KO cell line; however, this experiment was not conducted.

We absolutely understand why the reviewer has made this suggestion but unfortunately there is no commercially-available antibody against the COA5 protein that we have tried that would work to allow the detection of its submitochondrial localisation via SDS-PAGE. Given the small size of the protein at 12 kDa, it is difficult to determine its submitochondrial localisation unless a HA- or FLAG-tagged COA5 construct is generated and detected using antibody against the tag. This current manuscript focuses on the implications of dysfunctional COA5 protein on CIV assembly and therefore we would politely suggest that the investigation of the submitochondrial localisation of the mutant COA5 protein is outside of the immediate scope of this study, conceding that this would be an interesting experiment to conduct at a later point in time.

2. In Figure 2C, the measurement of activities is missing standard deviations (SD) for patient fibroblasts. Additionally, while it appears that complex III (CIII) is increased, the statistical significance is not indicated. Please include a notation for non-significant results and specify the significance of the other findings.

The assessment of OXPHOS enzyme activities in patient fibroblasts was only conducted once (activities are measured in triplicate, with different amounts of mitochondrial protein, and data presented as the mean of these three measurements) as part of an established diagnostic protocol (as exemplified in PMID: 31866046, 32969598) and therefore statistical significance could not be determined.

3. In Figures 3A and 3C, the Western blot analysis of patient samples shows significant differences only in CIV subunits. However, only MTCO2 was investigated. Quantification is needed, as the figures also suggest changes in a CI subunit (NDUFB8) and in UQCRC2 (increased in one instance and decreased in another).

Affected CIV steady-state levels were only represented by MTCO2 using the OXPHOS cocktail antibody (ab110411, Abcam) for a general overview of the OXPHOS protein steady-state levels in patient samples. The individual CIV subunits were then further investigated using the KO cell line (Fig 4A) due to the slower growth of patient-derived primary fibroblasts.

We thank this reviewer for their helpful suggestion. The quantification of western blot data has now been included for Fig EV1. Quantification of the signal intensity of each protein detected has confirmed the decrease in CIV subunit, MTCO2 and impaired CIV assembly examined through antibody against MTCO1. Both NDUFB8 and UQCRC2 signals appeared

to be at higher steady-state levels compared to controls. However, no statistical significance was established across biological triplicates for fibroblasts samples whereas skeletal muscle samples have only been tested once due to very limited availability of patient tissue and therefore we are unable to determine statistical significance for these data.

4. For the U2OS KO cell line, the authors should provide evidence that reintroducing the COA5 gene restores the KO phenotype. Additionally, they should demonstrate the absence of COA5 protein via WB, assuming the antibody is available and functional.

As previously mentioned in an earlier response to another point raised, there is no working antibody against the COA5 protein to study this by immunoblotting; however, the complete absence of COA5 protein in the COA5^{KO} cell line was successfully verified in MS-complexome profiling (**Fig. 6A**). Furthermore, the knockout cell line also evidently replicated the decreased MTCO2 steady-state level and impaired CIV assembly observed in the patient fibroblasts, affirming the COA5 knockout as the cause of isolated CIV deficiency, and thus in our view, not necessitating functional complementation experiments.

5. In Figure 4B, the authors should explain the additional band observed in UQCRC2 labeling in the KO cells.

The additional band observed in UQCRC2 has been consistently observed through the antibody used (ab14745, abcam) and could also be detected in wildtype cells when overexposed (**Figure R3, see below**). The additional, non-specific band was also indicated on the webpage of the vendor's site for this particular antibody (<https://www.abcam.com/en-us/products/primary-antibodies/uqcrc2-antibody-13g12af12bb11-ab14745#overlay=images>).

Figure R3 Corresponding to UQCRC2 panel in Fig. 4B in the manuscript. Increasing contrast on the same panel indicated additional non-specific band (red arrow) below the expected band in both wildtype and COA5^{KO} cell line.

6. Figures 4A, 4B, and 4C require better loading controls and quantification to assess the phenotype accurately. Suitable controls include Actin, VDAC, or TOM20.

We thank the reviewer for pointing this out. The loading control used was chosen with consideration of the molecular weights of proteins that were assessed, therefore rendering beta-actin, VDAC and TOM20 – all with similar molecular weights to the proteins of interest – inappropriate proteins to detect as loading controls. Quantification for Fig 4 has now been included. We would like to highlight the higher sensitivity of complexome profiling compared to other biochemical characterisation approaches (SDS-PAGE and BN-PAGE), hence the complexomic data provides the most robust verification in terms of protein abundance quantification in individual subunits as well as in complexes.

7. Figure 5 presents complexome profiling data that appear inconsistent with earlier statements about changes in steady-state levels of CIV subunits. The authors claim differences in I + III2 supercomplexes, but quantification comparing WT and KO is needed to assess increased or decreased levels accurately.

Thank you for this comment. Different solubilisation conditions were applied to the protein extracts in the complexome profiling, SDS-PAGE and BN-PAGE. Hence, the different biochemical experiments were always intended to assess relevant aspects of OXPHOS protein complex function/status depending on the respective protein extraction and solubilisation methods.

Most importantly, the isolated implication of COA5 knockout on MT-CO2 protein as shown in SDS-PAGE (**Fig 4A&B**) was replicated in the complexome profile of COA5^{KO} cell lines (**Fig 6**) while the MTCO1 and COX4 were unaffected.

The accumulation of I+III₂ supercomplexes (S₀) was deduced from the line graphs in **Fig 5** which showed that the relative abundance of CI and CIII were, in fact, increased by almost two-fold which has also been indicated by the heat map above, suggesting increased abundance of the S₀ supercomplexes. For better illustration, the heat map for averaged CI and CIII protein abundance has been included in **Fig 6A**.

8. The authors assert that COX1 steady-state levels are unaffected, yet no supporting evidence (e.g., WB) is provided.

Our assertion that MTCO1 steady-state levels were unaffected is based on data shown in **Fig 4A** where the COA5^{KO} cell line displayed a similar level of MTCO1 to the isogenic control at steady-state; this is now also verified through the quantification of the relative signal intensity (**Fig EV3**).

9. The discussion section repeats well-established observations about the stability of COX1 and COX2 in subcomplexes, the assembly of CIV, and the absence of respirasomes without CIV. Furthermore, the investigation of COA5 as a potential copper chaperone has already been covered in previous research published by other groups.

We thank the reviewer for these comments, something that Reviewer #1 has also highlighted. The discussion has now been revised to focus on the findings of this study which pinpointed the involvement of COA5 protein CIV assembly.

We very much appreciate that the role of the yeast homologue of human COA5 protein, Pet191, has been investigated extensively in terms of its affiliation to complex IV stability and function. Notably, the current understanding of the human COA5 protein has been largely deduced from its yeast homologue and the presence of a twin CX₉C motif. The role of the human COA5 protein has only been partially determined and described through the study of the patient report by Huigsloot et al. (2011) in addition to reported evidence of the COA5 protein restoring CIV activity in a COX11 knockout background, linking it to copper metalation during CIV assembly by Nyvltova, *et al.* (2022).

Our study managed to establish the specific role of human COA5 protein by characterising the consequences of its knockout biochemically, allowing precise deduction of its essential role in the incorporation of MTCO2 module to MTCO1 module during CIV assembly process. Additionally, the application of complexome profiling in this study allowed more in-depth understanding of the resultant stalling of early CIV assembly due to loss of COA5, serving as a powerful dataset to investigate the linkage of CIV assembly to the respiratory chain as well as the supercomplexes.

Reviewer #3 (Comments to the Authors (Required)):

The manuscript by Tang et al. studies a patient case with severe neonatal-onset metabolic disease and cardiomyopathy. Exome sequencing identified a previously described homozygous missense variant in the COA5 gene, which has been suggested to encode a complex IV assembly factor with thus far unknown mechanism of action. The authors utilize patient fibroblasts and muscle biopsy to show isolated CIV deficiency. They then go on to study CIV assembly in the fibroblasts and muscle biopsy using Western blot and BN-PAGE, which show a specific assembly defect. To aid the studies of CIV assembly, the authors generate a COA5 knock-out cell line using CRISPR/Cas9 and utilize this model to do mass spectrometric profiling of OXPHOS complexes, which further elaborates the role of COA5. The manuscript is well written, technically credible and all conclusion are supported by the experimental data. The only part that I find a bit puzzling is the claim that primary and immortalized patient fibroblasts grew poorly, and therefore generation of a knock-out cancer cell line was necessary. This is quite surprising because e.g. mtDNA-less rho zero cells can grow without the entire respiratory chain when uridine is supplemented. How were the fibroblasts immortalized? The patient cells harbor a COA5 missense mutation but the genome-edited cell line carries a full knock-out allele, yet the authors do not mention if it has a growth defect. The authors should either explain/clarify this and/or provide growth curves for the knock-out and control cell lines.

Dr. Jukka Kallijärvi

We are grateful to this reviewer for the time they have given to review and assess our manuscript and for their helpful comments.

We would like to highlight that the patient fibroblasts used in the experiments are not immortalised and therefore subject to senescence over time. On the other hand, the COA5^{KO} cells were generated using the immortalised U2OS cell line which should explain the relatively unaffected growth rate compared to patient-derived primary fibroblasts despite the loss of functional COA5 protein.

We include additional data for the reviewer to see in the form of **Figure R4** (see below), which represents the growth curve of the COA5^{KO} and isogenic control cell lines in glucose versus galactose media. It is evident that the COA5^{KO} cell line was unable to grow and survive in galactose medium when forced to rely on oxidative phosphorylation as sole energy source despite displaying feasible growth in glucose medium, albeit slower than the isogenic control.

Figure R4 Growth curve showing cell confluency in percentage against time in hours of COA5^{KO} and isogenic control cell lines. Cells were grown in glucose and galactose media over a span of 14 days.

December 11, 2024

RE: Life Science Alliance Manuscript #LSA-2024-03013-TR

Prof. Robert W. Taylor
Newcastle University
Institute of Neuroscience
4th flr. Catherine Cookson Building
Medical School
Newcastle upon Tyne, Tyne and Wear NE2 4HH
United Kingdom

Dear Dr. Taylor,

Thank you for submitting your revised manuscript entitled "COA5 has an essential role in the early stage of mitochondrial complex IV assembly". We would be happy to publish your paper in Life Science Alliance pending final revisions necessary to meet our formatting guidelines.

- please be sure that the authorship listing and order is correct
- please add the Twitter handle of your host institute/organization as well as your own or/and one of the authors in our system
- please rename your EV figures as supplementary figures and update all figure callouts in the main manuscript text accordingly
- please include a declaration indicating that written informed consent was obtained for the use of the patient samples

A. FINAL FILES:

B. MANUSCRIPT ORGANIZATION AND FORMATTING:

Sincerely,

Reviewer #1 (Comments to the Authors (Required)):

The authors have satisfactorily addressed all my comments/questions and in my opinion, this revised version of the manuscript should be accepted for publication.

Reviewer #2 (Comments to the Authors (Required)):

The manuscript has been now improved and most specific questions have been answered. However, a major issue remaining for this reviewer is the overall lack of novelty. It is mainly a confirmatory study.

*Specific comment: It has been reported in paper PMID: 35750769 the use of a working antibody against COA5 (PET191). This is HPA057768 from Sigma, at a dilution 1:500.

Reviewer #2 (Comments to the Authors (Required)):

The manuscript has been now improved and most specific questions have been answered. However, a major issue remaining for this reviewer is the overall lack of novelty. It is mainly a confirmatory study.

*Specific comment: It has been reported in paper PMID: 35750769 the use of a working antibody against COA5 (PET191). This is HPA057768 from Sigma, at a dilution 1:500.

Author's response:

Thank you for providing this information. We attempted to use the same antibody you describe, purchased through ThermoFisher (PA5-63480), but failed to detect COA5 protein consistently both in primary patient fibroblasts and immortalised U2OS cell lines. This has been reported to the manufacturer, hence their recommendation for the use of the antibody has now been removed from the ThermoFisher's webpage (<https://www.thermofisher.com/antibody/product/COA5-Antibody-Polyclonal/PA5-63480>).

December 19, 2024

RE: Life Science Alliance Manuscript #LSA-2024-03013-TRR

Prof. Robert W. Taylor
Newcastle University
Institute of Neuroscience
4th flr. Catherine Cookson Building
Medical School
Newcastle upon Tyne, Tyne and Wear NE2 4HH
United Kingdom

Dear Dr. Taylor,

Thank you for submitting your Research Article entitled "COA5 has an essential role in the early stage of mitochondrial complex IV assembly". It is a pleasure to let you know that your manuscript is now accepted for publication in Life Science Alliance. Congratulations on this interesting work.

DISTRIBUTION OF MATERIALS:

Again, congratulations on a very nice paper. I hope you found the review process to be constructive and are pleased with how the manuscript was handled editorially. We look forward to future exciting submissions from your lab.

Sincerely,
